# Bidirectional Interactions between the Menstrual Cycle, Exercise Training, and Macronutrient Intake in Women: A Review

**DOI:** 10.3390/nu13020438

**Published:** 2021-01-29

**Authors:** Sílvia Rocha-Rodrigues, Mónica Sousa, Patrícia Lourenço Reis, César Leão, Beatriz Cardoso-Marinho, Marta Massada, José Afonso

**Affiliations:** 1Escola Superior de Desporto e Lazer, Instituto Politécnico de Viana do Castelo, Rua Escola Industrial e Comercial de Nun’Álvares, 4900-347 Viana do Castelo, Portugal; ces.leao@gmail.com; 2Research Centre in Sports Sciences, Health Sciences and Human Development (CIDESD), Quinta de Prados, Edifício Ciências de Desporto, 5001-801 Vila Real, Portugal; 3Tumor & Microenvironment Interactions Group, i3S, Rua Alfredo Allen, 208 4200-135 Porto, Portugal; 4Nutrition & Metabolism, NOVA Medical School, Faculdade de Ciências Médicas, Universidade NOVA de Lisboa, Campo dos Mártires da Pátria, 1169-056 Lisboa, Portugal; monica.sousa@nms.unl.pt; 5CINTESIS, NOVA Medical School, NMS, Universidade Nova de Lisboa, Campo dos Mártires da Pátria, 1169-056 Lisboa, Portugal; 6Hospital da Luz Arrábida, Luz Saúde, Praceta Henrique Moreira, 150, 4400-346 Vila Nova de Gaia, Portugal; patricialoureis@hotmail.com; 7Oporto Sports Medicine Center, R. António Pinto Machado, 4100-068 Porto, Portugal; beatriz.marinho@fpf.pt; 8Centre for Research, Education, Innovation and Intervention in Sport, Faculty of Sport of the University of Porto. Rua Dr. Plácido Costa, 91, 4200-450 Porto, Portugal; martamassada@gmail.com (M.M.); jneves@fade.up.pt (J.A.); 9St. Mary’s Hospital of Porto, Rua de Camões, 906, 4049-025 Porto, Portugal

**Keywords:** women, sex hormones, menstrual cycle, exercise performance, nutritional intake, macronutrients, interindividual variability

## Abstract

Women have a number of specificities that differentiate them from men. In particular, the role of sex steroid hormones and the menstrual cycle (MC) significantly impact women’s physiology. The literature has shown nonlinear relationships between MC, exercise, and nutritional intake. Notably, these relationships are bidirectional and less straightforward than one would suppose. For example, the theoretical implications of the MC’s phases on exercise performance do not always translate into relevant practical effects. There is often a disconnect between internal measures (e.g., levels of hormone concentrations) and external performance. Furthermore, it is not entirely clear how nutritional intake varies across the MC’s phases and whether these variations impact on exercise performance. Therefore, a thorough review of the existing knowledge could help in framing these complex relationships and potentially contribute to the optimization of exercise prescription and nutritional intake according to the naturally occurring phases of the MC. Throughout this review, an emerging trend is the lack of generalizability and the need to individualize interventions, since the consequences of the MC’s phases and their relationships with exercise and nutritional intake seem to vary greatly from person to person. In this sense, average data are probably not relevant and could potentially be misleading.

## 1. Introduction

The biological differences between men and women contribute to several sex-specific features. Many of those differences are largely due to sex steroid hormone fluctuations, especially estrogens and progestogens [1]. Both endogenous estrogens and progestogens fluctuate predictably across the menstrual cycle (MC) in naturally eumenorrheic women [2]. Beyond reproductive function, both hormones have a huge impact on many tissues, including skeletal muscle, cardiac tissue, bone, connective tissues, and the central and peripheral nervous system, among others [3,4,5]. An important question that is unexplored is whether exercise training changes the endogenous production of these two hormones during the MC, or if there is an adaption to the demands of training and/or competition [6,7].

As we will attempt to establish, the most up-to-date evidence supports the notion that a complex and bidirectional relationship exists between MC and exercise training, with reciprocal influences and adaptations as well as important inter- and intra-individual variations [8,9]. For example, one study reported that alterations in strength, metabolism, body temperature, fluid balance and injury risk are concomitant with hormonal fluctuations throughout the MC and can affect approximately 75% of athletic women [10], pointing out that the way which MC affects exercise performance is highly individual. In addition, both the MC and the responses to exercise training may be mediated by nutritional status. Furthermore, nonlinear dose-response relationships suggest the adoption of equally nonlinear periodization and programming strategies to better account for inter- and intraindividual variation, establishing an ongoing dialogue between plan and process instead of relying on pre-determined plans [11].

In this vein, the relevance of individualized dietary advice in women is becoming increasingly recognized, with dietary strategies varying according to women’s physiology, especially due to the different sexual hormone concentrations during the MC [12,13,14]. Indeed, levels of endogenous estrogens and progesterone affect the proportions of macronutrients used as fuel not only at rest, but also during exercise training. To counteract these imbalances, an adjustment in nutritional intervention to the MC phase in eumenorrheic women may be required. Accumulated evidence has suggested that estrogens improve exercise performance by varying carbohydrate (CHO), lipids and protein metabolism, although progesterone commonly acts antagonistically [15,16,17]. Despite inconsistent findings being reported, a thorough knowledge of complex and bidirectional interaction between MC, exercise training and macronutrient intake is highly important to further optimize exercise training effects in athletic women, in both health and exercise performance contexts.

Therefore, in the present review, we focused on the current evidence of the complex and bidirectional interactions between MC, exercise training and macronutrient intake in both health and exercise performance contexts, providing an overview of the effects of exercise training and macronutrient intake in the underlying MC-related mechanisms in women. Briefly, in the first part of this review, we will examine the bidirectional interaction of endogenous estrogens and progestogens as well as MC phases in women’s physiology and exercise performance. In the second part of this review, we will explore the reciprocal interaction of energy availability, macronutrient intake and MC phases in pre-menopausal women. Two special boxes will address issues pertaining to the menopause and the intake of combined oral contraceptives.

## 2. State of Art

### 2.1. Bidirectional Relationships between Exercise Training and Sex Steroid Hormones

There are six families of steroid hormones: androgens (e.g., testosterone, androstenedione), estrogens (e.g., estradiol, estrone), progestogens (e.g., progesterone, 17-hydroxyprogesterone), glucocorticoids (e.g., cortisol), mineralocorticoids (e.g., aldosterone), and vitamin D [2,3]. In the present review, we will focus on estrogens and progestogens. Together with androgens, these are classified as sex steroid hormones [1]. As sex steroids, these hormones play a crucial role in physiological functions, including reproduction, mainly by the hypothalamic–pituitary–gonadal axis, sexual differentiation (secondary sex characteristics) and sexual behavior patterns as well as metabolic processes in adipose tissue, skeletal muscle and connective tissues, among others [3,4,5].

Women have a 20- to 25-fold lower circulating concentrations of androgens compared with men [18]. As precursors of estrogens synthesis, androgens play a key role in the maturation processes of ovarian follicles in women [19]. However, little is known about the interaction of the androgens and exercise training in women [20]. Although the additional biological role of testosterone in women remains unclear, a therapeutic role for androgens in the functional limitations of ageing has arisen [20], but this is out of the scope of this review.

Circulating estrogen and progestogen concentrations induce well-known effects in cardiovascular [21,22], respiratory [23] and metabolic processes [24,25] with subsequent implications for strength and aerobic and anaerobic performance. However, much less is known about the effects of androgens in women engaging in exercise [26]. Thus, the sex steroid-related hormonal pathway’s response to exercise training in health and exercise performance contexts should be reviewed.

#### 2.1.1. Estrogens

Estrogens are a class of sex steroid molecules produced from cholesterol and secreted by the ovaries and placenta [2,25], of which the three major forms in women are estrone (E1), estradiol (E2) and estriol (E3) [27]. While circulating in the blood, estradiol is bound to a protein carrier, known as sex hormone binding globulin (SHBG), which is produced in the liver; another 30% is loosely bound to albumin, leaving only about 1% unbound and free. The biologic effects of the major sex steroids are largely determined by the unbound and free hormones [19]. Estrogens are primarily involved in the development and maintenance of normal sexual and reproductive function in women [4,5].

In women, circulating estrogens result from the ovarian secretion of estradiol and estrone, and peripheral conversion of its precursors occur in fat tissue, the skin, in muscle, and in the endometrium [19]. Estrogens fluctuate naturally throughout a woman’s life [3]. As steroid hormones, they act by binding to steroid receptors through a classical pathway where they regulate gene expression [3]. Additionally, sex steroid receptors can be found outside the nucleus, including in the mitochondria, the endoplasmic reticulum, and the plasma membrane, where they activate different signaling cascades, exerting their action through a nonclassical pathway [3].

As steroid hormones, estrogens can freely pass through the plasma membrane and move into the nucleus, binding to its nuclear receptors: classical nuclear estrogen receptors (ERα and β) and novel cell surface membrane receptors (GRP30 and ER-X) [3]. Therefore, the post-transcriptional effects of estrogens include cell redox state regulation [28,29], mitochondrial function [3] and direct interference with the activity of specific enzymes, such as aromatase [30]. For example, aromatase is the enzyme responsible for the biosynthesis of estradiol and estrone from androgens, while, in postmenopausal women, this reaction usually occurs in white adipose tissue, where aromatase activity is augmented [30].

The accumulated evidence has shown that exercise training has a substantial impact on several female sex steroid-related hormonal pathways in both health and exercise performance contexts [6,7]. As recently reviewed by Rocha-Rodrigues [31], some mechanisms have been proposed for the protective role of exercise training through estrogen-mediated signaling in improving health, such as cancer and the menopause (which will be the focus of a special box). Randomized controlled trials (RCTs) conducted among healthy women demonstrated a significant decrease in total and free circulating estradiol concentrations induced by physical activity [32,33]. A clinical study [34] involving sedentary premenopausal women reported that a four-cycle intervention of moderate-intensity aerobic exercise combined with energy restriction resulted in significant decreases in serum estradiol and urinary estrone-1-glucuronide and pregnanediol glucuronide levels.

In contrast, Smith et al. [35] found no alterations in estradiol, estrone or SHBG levels after 16 weeks (4 MCs) of aerobic exercise (150 min per week at 65–70% of the maximum age-predicted heart rate) in sedentary, healthy, young and eumenorrheic women. Although studies demonstrated the effect of exercise training on free estradiol, a small part of those reporting on total estradiol revealed that the effect of exercise training was more obvious for free estradiol than for total estradiol [36,37]. As reported by Ennour-Idrissi et al. [36], total estradiol levels decreased in response to intervention-induced weight loss, but the low levels of free estradiol observed were not attributed only to exercise training-mediated weight loss, but also to increased levels of binding proteins, as in the case of SHBG levels. These findings are interesting, as increased SHBG was associated with decreased amounts of unbound, active estrogens and androgens in circulation, which have been linked to a protective phenotype in women [38].

Estrogens exert important biological functions in the development, maturation and aging in extragonadal tissues, including in the bone [39], cardiac [28,29] and skeletal muscles [40], as well as connective tissues [41]. It is important to note that estrogen may play distinct roles in different tissues, mainly owing to intricate crosstalk between circulating and tissue-specific estrogens; therefore, their beneficial or harmful effects remain under debate. The powerful antioxidant capacity of estrogen to protect cardiac muscle [28,29] through its antioxidant and membrane-stabilizing properties is well documented, but this effect in skeletal muscle is not completely understood. Recent reports have suggested that estrogen has a noticeably positive effect on skeletal muscle mass and strength, albeit in animal models [40,42,43], suggesting that in the absence of estrogen, skeletal muscle is more prone to injury, thereby limiting regrowth. Therefore, it is hypothesized that estrogen could stabilize the extracellular matrix or act as an antioxidant to decrease muscle injury; however, the physiological significance of this effect on human skeletal muscle has not been clearly defined because of estrogen fluctuations and/or confounding factors such as age, fitness levels, exercise type and/or intensity.

There is evidence to support that both estrogens and ERs play a crucial role in the musculoskeletal tissues, such as the bone, ligaments, and tendons through regulation of CHO and lipid metabolism. Moreover, the data support the hypothesis that the ablation of ERα in skeletal muscle results in muscle mass loss, suggesting that the beneficial effects of estradiol on muscle strength might be receptor dependent [42]. In vitro and in vivo assays of skeletal muscle-specific ERα knockout mice showed that muscle contractility was impaired. These results support the hypothesis that a primary mechanism through which estradiol elicits its effects on strength is mediated by ERα. Evidence has been presented in support of the belief that estradiol signaling through ERα appears to modulate force at the molecular level via posttranslational modifications of the myosin regulatory light chain.

A controversial topic concerns the hypothetic estrogen-mediated mechanism concerning ligament laxity. In fact, ERs are expressed in the anterior cruciate ligament (ACL) and the inhibitory effect of estrogen on fibroblast proliferation and collagen production in human ACL has been reported [44]. From a clinical point of view, alterations in ACL in association with estrogen concentrations likely provides it with a greater susceptibility to injury. This issue will be explored further in Section 2.2.

In summary, the mechanisms by which estrogens interact with exercise performance in women is still unclear. There are complex mechanisms of action at play; therefore, more research is needed in order to explore the pathways by which estrogens could act on skeletal muscle in premenopausal and postmenopausal women. Furthermore, it is highly possible that considerable interindividual variation exists, which will also be analyzed in Section 2.2.

#### 2.1.2. Progestogens

Progesterone is the major and most important progestogen in the body, and it is an endogenous steroid hormone primarily synthesized by the ovarian corpus luteum (which produces the majority of progesterone), adrenal and mammary glands [2,45]. Progesterone has functions in maintaining pregnancy (secreted by the ovarian corpus luteum during the first 10 weeks of pregnancy and by the placenta in the later phase of pregnancy), but also in other phases of the MC [19]. Moreover, progesterone has several biological activities in the human body, especially within the reproductive system, including the facilitation of implantation, and the maintenance of pregnancy due to its promotion of uterine growth and suppression of myometrial contractility, as well as the development of functional milk-producing alveolar lobules in the mammary gland [45].

Primarily recognized as a hormone of the reproductive system, progesterone also plays a functional role in the neuroendocrine axis, as well as in the musculoskeletal system, in both men and women, throughout life [46,47]. Its neuroprotective role has been demonstrated in both the central and peripheral nervous system, influencing the control of myelination and myelin plasticity by astrocytes [48]. Generally, progesterone likely interacts with estrogen to induce concerted functional and metabolic effects [49]; however, some effects can antagonize each other’s effects [50]. The isolated effect of progesterone on skeletal muscle function and growth has been poorly described [51], as the roles of progesterone receptors—commonly found as three isoforms of progesterone (PR-A, PR-B, and PR-C)—in skeletal muscle are not as clear as those of estrogen receptors [52]. In fact, Greeves, Cable, and Reilly [53] found that increased quadriceps strength was associated with progesterone concentration. However, Janse De Jonge et al. [51] found that skeletal muscle contractile properties (e.g., isometric quadriceps strength with superimposed electrical stimulation) were not affected by the fluctuations in progesterone levels throughout the MC in healthy women with regular MCs. These observations have been reported by others [54,55].

Furthermore, treatment with progesterone seems to have no effect on the maximal activity of several key enzymes of lipid oxidation in both red and white rat skeletal muscle cells [56]. In the same study, the combination of estradiol and progesterone induced similar fat oxidation enzyme activities to those of ovariectomized rats, demonstrating that progesterone inhibited estradiol in physiological concentrations. Similarly, other studies found that progesterone antagonizes the stimulation of hepatic triglyceride secretion [57] and fatty acid oxidation [15] induced by estradiol in nonovariectomized female rats. Biochemically, the anti-estrogenic actions of progesterone (and also progestins) in women is mediated by the synergetic actions of estradiol (E2) and progesterone, in which a decrease in the estradiol (E2) receptor along with the synthesis of 17-hydroxysteroid dehydrogenase result in an increase in the conversion of estradiol (E2) into a less active estrone (E1) in the target tissues [58,59].

Despite the high level of interest in the effects of the female sex steroid hormones on both health and exercise performance, there remains considerable controversy in the literature regarding the effects of exercise training on progesterone. One study [60] found no changes in progesterone levels after incremental exercise on a cycle ergometer in top five basketball players, whereas others [61] reported progressive increases in progesterone levels with an incremental exercise in the luteal phase (LP) in four teenage swimmers. In healthy and untrained women (*n* = 10), two hours of running at 70% of maximal oxygen consumption (V̇O_2max_) decreased progesterone levels but increased the metabolic clearance rate of this hormone [62]. In fact, during moderate-to-high intensity exercise training, with a generalized increase in body metabolism, the metabolic clearance rate of progesterone (and also estrogen) increases, which may contribute to the decrease in hormone levels. Nonetheless, the degradation of sex steroid hormones depends on their metabolism in the splanchnic area, which decreases during high-intensity exercise training sessions, thus resulting in increased post-exercise sex steroid hormone levels [63].

#### 2.1.3. Estrogens, Progesterone and Exercise Training

Mechanisms have been proposed by which both ovarian sex steroid hormones, estrogen and progesterone, affect women’s physiology and consequently their exercise performance [17]. Specifically, estrogens are thought to have an anabolic effect on skeletal muscle [64] with a crucial role in substrate metabolism changes through increased muscle glycogen storage capacity, free fatty acid availability and the use of oxidative pathways [15,17,64]. Therefore, this mechanism decreases the dependence on anaerobic pathways for ATP production, and consequently lowers blood lactate levels, thus resulting in less fatigue [16]. This metabolic hormonal action perhaps contributes to improving the ultra-endurance exercise capacity in women vs. men. It should be noted that this oxidative energy-dependent pathway may occur at certain specific exercise intensities, and at a higher relative effort, dependent, of course, on increased blood glucose and muscle glycogen stores [16,64].

In turn, endogenous progesterone and the synthetic progestins have a central thermogenic effect that is responsible for the increase in the basal body temperature in LP of the cycle [65]. Hence, an increase in basal body temperature is reported to increase the subjective feeling of greater exertion or strain when exercising, decreasing athletic performance, especially in hot and/or humid environments [66]. In the same study, progesterone induced an increased ventilation and maximal exercise response during the LP of the MC. As mentioned above, progesterone has the ability to antagonize estrogen actions and, thus, high levels of progesterone can constrain the increased CHO metabolism induced by estradiol [67]. Known as the catabolic breakdown effect hormone, progesterone can also reduce the muscle protein synthesis [68].

It should be noted that, during exercise training, ovarian sex steroid hormones may have indirect effects on substrate metabolism through interactions with other hormones, particularly catecholamines [69]. These findings suggest that the fluctuations in ovarian sex steroid hormones during the MC have the potential to interfere with exercise performance in women.

### 2.2. Bidirectional Relationships between Exercise Training and the Menstrual Cycle

After the menarche, women experience MCs, i.e., naturally occurring, hormone-dependent cycles deeply related to the female reproductive system, and in particular to levels of estradiol, progesterone, follicle-stimulating hormone (FSH) and luteinizing hormones (LH) [16,70]. In some cases, hormonal secretion can vary from 10- to 100-fold during the MC [47]. The nonpathological MC can vary between 26 and 35 days [71], and is associated with variations in sex hormone levels, which have been hypothesized to affect neuromuscular performance and the likelihood of musculoskeletal injury [16,72,73]. In the study of Ekenros et al. [72], fifteen healthy women with regular MCs volunteered for biopsies from the vastus lateralis at three verified time points: follicular phase (FP), ovulatory phase (OP) and LP. During the MC, significant variation was found for mRNA and protein levels of estrogen ERα, which were highest in FP, while progesterone levels were highest in the LP. Incidentally, no significant fluctuations were found for androgen receptors. The authors postulated that these fluctuations may have an impact on responses to exercise training and on the risk of injury, but these were not assessed during the study, and therefore remain speculative.

In a study with fifteen eumenorrheic sedentary women aged 22.1 ± 1.0 years, not taking oral contraceptive drugs for at least 6 months prior to the experiment [74], the participants performed exercise on a cycle ergometer at 60% of V̇O_2max_ during 45 min, followed by exercise of progressive intensity until 80% of V̇O_2max_ until exhaustion. The cross-over protocol was conducted in the early FP and also in the mid LP, after a 6-month period of monitoring the MC and basal body temperature every morning of the previous 2 months. In the LP period, women had significantly lower serum total carnitine and free carnitine, but blood levels of estradiol, progesterone and acylcarnitine were not significantly different. Interestingly, one group had superior endurance performance in the LP, while the other group had superior performance in the FP. Therefore, and against the authors’ conclusions, data showed differences in only two of the five blood markers, and this was not correlated with endurance performance. In a similar vein, a narrative review speculated that MCs could increase the production of hepcidin, preventing macrophages from releasing iron and reducing the intestinal absorption of dietary iron, thereby interfering with iron regulation [75]. However, the authors concluded that these relationships were still unclear, and the relationships with exercise performance were entirely hypothetical.

A systematic review with a meta-analysis (SRMA) of 21 studies and 68,758 participants aimed to identify the relationships between the MC phase and the utilization of oral contraceptives on the laxity of the ACL and noncontact injuries [76]. The data suggested a link between the phases of the MC and the likelihood of a noncontact ACL injury, and also that oral contraceptives could reduce this risk by up to 20%. However, the authors emphasized the low quality of the evidence, advising against more definitive conclusions. Another systematic review including 17 studies [77] showed increased ACL injury risk during the pre-OP, especially in women with ACL laxity, due to kinematically altered patterns, as assessed through functional activities, including greater dynamic knee valgus and external rotation of the tibiae.

A study of 13 women with regular MCs, eight of which had premenstrual syndrome (PMS), showed that women with PMS had greater postural sway and greater threshold for detecting passive knee motion than women without PMS [78]. The authors of this study speculated that this could explain the increased incidence of exercise-related injuries in the LP. Postural sway in a static standing posture was also assessed in 18 healthy women (19.11 ± 0.9 years-old) [79]. The task was performed 1–3 days after menstruation and repeated 13 days after menstruation. In the second moment, velocity moments of postural sway were significantly higher, suggesting that the MC affects the static balance of healthy women. The authors further speculate that balance exercises could therefore relate to injury prevention. However, this study presented no investigation or demonstration that balance training could alter this MC-related dynamic in static balance. In line with the small sample size and the absence of the calculation of effect sizes, the authors’ conclusions may have been premature. Unfortunately, unsupported speculation is common in this field of research. A study with 30 sedentary healthy women aged between 18 and 25 years old with a regular MC [80] assessed static and dynamic balance and stated that both had better scores in the OP in comparison to early FP, and on that basis suggested the inclusion of balance-based exercise programs. However, no effect sizes were reported, and the effectiveness of such programs in changing these fluctuations over the course of the MC were not assessed.

Research has further explored the relationships between the MC’s phases and flexibility. In a study with 20 women aged between 18 and 35 years old, engaged in gymnastics classes at fitness centers, and not taking oral contraceptives, Melegario, Simão, Vale, Batista, and Novaes [81] assessed flexibility at three points in time (FP, LP and OP) using goniometry to assess eight movements across five joints (shoulder, elbow, lumbar column, hip, knee). There were no significant differences in flexibility across the three phases. Consistent with these findings, a study with 20 women not using hormonal contraceptives (HCs) and 24 women using HCs [73], flexibility was assessed in FP, OP and LP using the sit-and-reach test and, again, no significant differences were found between groups and between MC phases. Therefore, the hormonal fluctuations associated with the MC do not seem to alter flexibility levels in young, healthy women.

Beyond objective measures, however, it may also be relevant to consider the perceptions of athletic women with regard to the relationships between MC and sport performance. Interviews with fifteen international rugby players (24.5 ± 6.2 years) showed that >90% of athletes reported symptoms related to the MC [82]. Almost 70% reported that the symptoms during menstruation interfered negatively with their performance. Regardless of whether this effect on performance was real or merely perceived, it may still lead to important challenges for the athlete and the technical staff. For example, in the above-mentioned study with rugby players, common symptoms during menstruation included reduced energy levels, worry, distraction, impaired motivation and fluctuating emotions.

An RCT with healthy but sedentary women aged between 18 and 45 years-old analyzed 73 women not taking hormonal contraceptives (HCs) and a control group taking HCs [83]. The women performed a treadmill exercise at 65% of V̇O_2max_ in different stages of the MC (or equivalent days, in the case of women undertaking HCs), after which perceived exertion and pain were self-reported using Borg’s Rating of Perceived Exertion (RPE) and the Borg Category Ratio-10 (CR-10), respectively. Women in the early FP not taking HCs presented with significantly greater increases in RPE and pain, in comparison with the late FP and LP. In a similar vein, a study with 9 elite and 21 non-elite athletic women monitored the assessed salivary testosterone levels before breakfast during the FP, OP and LP of the MC, followed by two questions related to competitive desire and training motivation [84]. The OP was associated with the highest concentrations of salivary testosterone, with a significantly more accentuated response in the elite group. This was accompanied by an equivalent increase in competitiveness ratings. Therefore, in this study, an objective and a subjective measure peaked during the same phase of the MC, and this relationship was stronger in elite athletes. However, training factors such as modality, intensity and volume were not monitored.

Establishing causal relationships is a complex venture, usually dependent on well-designed and implemented RCTs [85]. Furthermore, most interventions tend to be multimodal, which precludes a more thorough understanding of the effects of unimodal interventions and limits our understanding of physiological responses to a given exercise protocol [86]. Despite some evidence suggesting that estrogen may increase performance in endurance exercise through a change in macronutrient metabolism [16], perhaps it is naïve to expect that MCs interfere linearly with exercise performance. MCs promote hormonal and mechanical changes that could theoretically have an impact on exercise performance, but these hypothetical links are not necessarily backed up by research. Indeed, the relationships between MCs and exercise performance have presented contradictory evidence [16]. Often, hormonal fluctuations do not reflect changes in muscle contractility, lactate kinetics, bodyweight or heart rate, among other relevant variables [9].

Additionally, *p*-values only inform about the probabilities of an event, but it would be relevant to understand the magnitude of the effects, which is precisely what Pereira, Larson, and Bemben [87] attempted to do in a recent review. Analyzing studies with eumenorrheic women, the authors included 46 studies reporting on motor output in the FP and LP of the MC. Only 15 of the 46 studies showed statistically significant differences between FP and LP, which, again, denotes that MC-related hormonal changes do not translate linearly into assessments of motor performance. Equally importantly, effect sizes varied widely in terms of magnitude and direction, i.e., they expressed both quantitative and qualitative differences. The authors present potential confounding factors, such as upper versus lower limbs, isometric versus dynamic contractions, single limbs versus full body, as well as how the phase of the MC was assessed. However, in our opinion, the results highlighted by this review reinforce the notion that there may be high interindividual variability in how the MC affects exercise performance. Moreover, adherence rates may influence the effects of any exercise program, and so this factor should also be considered [86].

In this vein, a recent SRMA analyzed the relationships between strength-related variables and the phases of the MC in eumenorrheic women [88]. Twenty-one studies permitted a comparison between early FP, OP, and mid-LP in relation to isokinetic peak torque, explosive strength, and maximal voluntary contraction. Beyond the effects being nonsignificant, again, effect sizes varied in magnitude (i.e., quantitatively) and in direction (i.e., qualitatively), reflecting the uncertainty associated with the relationships between the MC phase and performance in strength tests. Notably, the authors reported a high risk of bias in the study design. In fact, the authors reported a high level of bias, but the assessments report a risk of bias that might or might not translate into actual bias. Therefore, in line with Cochrane’s recommendations, we prefer the term risk of bias [89]. Furthermore, Hayashida, Shimura, Sugama, Kanda, and Suzuki [90] found an increase in the levels of two inflammatory markers (IL-6 and calprotectin) in all three MC phases after 60 min of cycling at 75% of the anaerobic threshold. These findings suggest that exercise training at high intensity increases stress and inflammation regardless of the phase of the MC, even though the authors have concluded that this was more apparent in the menstrual phase. Conversely, low intensity exercise does not seem to have a large effect on inflammation or cell-mediated immune function [91].

The relationships between MCs and exercise performance should probably be analyzed bidirectionally, with reciprocal influences and adaptation [8,92], and subject to inter- and intraindividual variation [9]. These relationships may even be traced to before the establishment of regular MCs. Sports that promote a low body mass are associated with menstrual dysfunction and may also be associated with delayed puberty [93]. Does intensive exercise delay the menarche, however, or do girls with delayed menarche benefit from this in some types of exercises or sports? In general, this question remains up for debate. Through regular exercise, it is possible that women learn how to maximize their performance regardless of their MC phase, especially for high-intensity exercise bouts [9]; therefore, exercise performance may be minimally affected by the phases of the MC [88]. Moreover, energy demands and nutritional status should be considered important confounding variables [16], which is why they are the subject of Section 2.3.

This section can perhaps be best summarized through the findings of a recent SRMA assessing the effects of the MC phase on exercise performance in eumenorrheic women [94]. The authors found that exercise performance could only be trivially reduced during the early FP in comparison to the other phases of the MC. They further reported large between-study variations in responses and the overall poor quality of the studies, concluding that generic guidelines on exercise performance across the phases of the MC could not be established. Finally, the authors suggest a personalized approach, considering each woman’s response to exercise across the MC. We fully agree with the authors’ conclusions Box 1.

Box 1Special box—Menopause and exercise training.Menopause is a physiological process characterized by a reduction in circulating estrogen as an effect of the reduced sensitivity of the ovary to circulating gonadotropins—follicle-stimulating hormone (FSH) and luteinizing hormone (LH)—caused by a significant decrease in available binding sites due to the reduction in follicle numbers [95]. Thus, this lower sensitivity results in decreased estrogen synthesis and increased anovulatory cycles. In postmenopausal women, daily physical activity levels were inversely associated with circulating concentrations of estradiol and estrogen, but positively associated with SHBG levels [96,97]. These observations were accompanied by weight loss and decreased abdominal adipose tissue mass [98,99], the main source of estrogen synthesis after menopause.Recently, a meta-analysis comprising 6 RCTs [100] has shown that combined intervention, including low calorie intake and exercise training, with durations ranging from 16 to 52 weeks, had a positive impact on estrone, total and free estradiol, and SHBG levels in healthy postmenopausal women. The increase in SHBG was also observed by McTiernan et al. [99] after 12-week moderate-intensity aerobic exercise, which likely resulted in decreased amounts of unbound, active estrogen. Similar effects were observed by van Gemert et al. [101] in response to 14 weeks of intensive combined aerobic and resistance exercises (4 h/week) with additional reduced fat mass and increased fat-free mass (lean mass). Taken together, these data suggest that exercise training was relatively effective in decreasing circulating estradiol levels independent of weight loss, which highlights the benefits of regular exercise training for women.In fact, a loss of estrogen may result in greater aging-related strength loss and a reduced rate of strength gain [102]. After menopause, the rapid decline in muscle mass may be explained by the increase in the protein synthesis rate, which is counteracted by a greater increase in protein breakdown or by the fact that the proteins synthesized are not myofibrillar proteins, but rather those needed for injury repair [47]. Furthermore, postmenopausal women show reduced sensitivity to anabolic stimuli when compared to age-matched men, which may suggest that a chronic decline in estrogen decreases the response to anabolic stimuli [103]. Myofibrillar protein synthesis in women taking estrogen replacement therapy (ERT) increased in response to strength training, but not in postmenopausal women who did not take ERT [104], which emphasize the role of estrogen in determining the sensitivity of the muscle to anabolic signaling. A well-designed study dealing with monozygotic twin pairs (*n* = 16) who were discordant for hormone-replacement therapy (HRT) use (one twin was on HRT while the other was not), the thigh muscle cross sectional area and relative muscle area were greater in the twins taking HRT than their sisters [105]. Similarly, Sipilä et al. [106] found that the muscle cross-sectional area and knee extension torque increased in exercised postmenopausal women taking HRT. Based on compelling data from studies in postmenopausal women, exercise training in combination with HRT is more effective in fostering skeletal muscle performance and mass than either HRT or exercise alone in postmenopausal women.However, the discussion is still ongoing. Recently, an SRMA [107] verified the effects of HRT for protection against the age-related loss of lean body mass in women aged 50 years and older. Only randomized trials were included. Twelve studies comprising 4474 participants afforded an analysis of 22 intervention arms using only estrogen, and 15 using estrogen and progesterone. Controls either received no HRT or a placebo. While 14 studies showed a protective effect of HRT with regard to not losing lean body mass, seven studies still showed a significant loss. More importantly, the difference between interventions and controls was not significant, and the absolute mean difference was of a mere 0.06 kg (95% CIs: −0.05 to 0.18) in favor of the interventions. Stratification based on the type and dosage of treatment, duration of follow-up, time since menopause, study quality and the specific protocol for assessing lean body mass provided similar results. The evidence was considered low quality based on GRADE. Therefore, perhaps there is still a considerable amount of work to be conducted within this field.

### 2.3. Bidirectional Relationships between Macronutrient Intake and Sex Steroid Hormones

The relevance of individualized dietary advice in women is becoming increasingly recognized, with dietary strategies varying according to the health status, physical condition and endogenous estrogen and progestogen variations during the MC [12,13,14,108]. Overall, the concentrations of estrogen and progesterone have an impact on the utilization of macronutrients not only at rest, but also during exercise. Therefore, there might be a need to adjust the nutritional interventions during the MC phase in eumenorrheic women, especially because nutritional habits may change during such phases [13].

Nevertheless, to the best of our knowledge, few studies have investigated the hypothetic impact of diet manipulation, e.g., the percentage of endogenous sex steroid hormonal levels over the course of the MC in relation to the total energy value of various macronutrients. Furthermore, the changes that are observed throughout the MC in terms of the metabolism of macronutrients, particularly the levels of circulating hormonal concentrations, will have implications for strategies that could potentially work to increase the performance of athletic women [16]. For this reason, there is a need to design study interventions that take these changes into account and that can help define which strategies may actually be successful. The proposed nutritional strategies should be applied according to the menstrual phase for eumenorrheic and pre-menopausal women.

#### 2.3.1. Energy

In order for all the systems to work properly, a sufficient amount of energy must be ingested. The energy availability is the difference between the energy intake and energy exercise expenditure in relation to fat-free mass (FFM) and is considered to be optimal for women when it is ≥45 kcal^−1^·kg^−1^·FFM^−1^·day [109]. A recent SRMA [110] found that the MC exerted a small but statistically significant effect on the resting metabolic rate (RMR) in women, with higher values during the LP compared to the FP. The difference between the two phases is estimated to be between 100 and 300 kcal [111]. Curiously, this increased energy expenditure seems to be naturally compensated by an increase in energy intake during the LP. This was shown by the study of Barr, Janelle, and Prior [112], where women consumed around 301 kcal more in the LP compared to the FP. This might be mediated by progesterone since this hormone can increase the appetite and food intake in the presence of estrogen [113]. In conclusion, the energy expenditure seems to be higher in the LP compared to the FP (100–300 kcal), and this difference seems to be naturally compensated by an increase in the energy intake during this phase of the MC.

#### 2.3.2. Carbohydrates

Women rely less on CHO oxidation to support fuel requirements compared to men [12]. Carter, Rennie, and Tarnopolsky [114] investigated the effect of 7 weeks of endurance training using a cycle ergometer on whole body substrate, glucose, and glycerol utilization during 90 min of exercise at 60% peak O_2_ consumption in men (*n* = 8, healthy) and women (*n* = 8, healthy, tested in the early to mid FP of the MC). Considering the lower respiratory exchange ratio, glucose metabolic clearance rate, glucose rate of appearance and disappearance, and the higher exercise glycerol rate of appearance and disappearance, the authors concluded that women oxidize proportionately more lipids and less CHO during exercise compared with men. In turn, women may favor the utilization of lipids in moderate-intensity exercise training of long duration, in comparison with men of equivalent training and nutrition status [115].

Additionally, exercise training-induced glycogen reduction is attenuated in women in comparison with men, especially in type I muscle fibers [116], and there are sex-related differences in the amounts necessary to obtain maximal glycogen reserves [108]. Devries et al. [12] observed that women (*n* = 13, healthy, recreationally active) in the LP, but not in the FP, used less glycogen during a 90 min bike ride at 65% peak oxygen uptake compared to men (*n* = 11). Additionally, the authors found that, in the LP, there was a lower glucose rate of appearance, rate of disappearance and metabolic clearance rate at 90 min of exercise compared with the FP. Estrogen leads to a decrease in muscle glycogen use in the LP compared to the FP, while also decreasing CHO use during exercise [12]. At rest, glucose levels were also lower at the FP compared to the LP, as shown by McLay et al. [14] in a study with nine moderately trained women. Due to this, and compared to men, an adjustment of the CHO intake might be postulated throughout the MC, with a decrease in the CHO requirements in the LP and possibly also in the FP. Moreover, estrogen seems to mediate a favorable effect on insulin sensitivity in women compared to men [117]. Due to this, an adjustment of the CHO intake might be postulated for athletic women throughout the MC, with a decrease in the CHO needs at rest and during exercise. These changes seem to be more pronounced in the LP, although they can also occur in the FP.

For CHO consumption during exercise, Wallis, Yeo, Blannin, and Jeukendrup [118] demonstrated that, in endurance-trained women (*n* = 8), the highest rates of exogenous CHO oxidation and greatest endogenous CHO sparing were observed when CHO was ingested at a rate of 60 g/h during a 2 h cycling exercise bout (≈60% V̇O_2max_), with no further increases at an ingestion rate of 90 g/h. The general recommendations for CHO intake during exercise are 60 g.h^−1^ for exercise with a 2–3 h duration and only 90 g·h^−1^ when the exercise event is >2.5 h [119]. Therefore, the results of Wallis et al. [118] are in line with general recommendations since, for a 2 h exercise event, the CHO intake recommendation would be 60 g·h^−1^.

Regarding CHO loading, an increase in energy intake (≈30%) might be needed to achieve a significant increase in CHO intake, at least in the FP. In a study [108] comparing the different energy levels for CHO requirements to maximize glycogen storage in both female (*n* = 7) and male (*n* = 6) endurance-trained athletes, the authors found sex-related differences in the required energy levels. Indeed, athletic women were able to significantly increase their muscle glycogen concentration only when the increase in CHO intake (from 5.1 g·kg^−1^·day^−1^ to 8.8 g·kg^−1^·day^−1^) was combined with a 34% increment of the total energy intake. In the trial where only CHO was increased, the total daily amount was 6.4 g·kg^−1^.day^−1^. In contrast, men endurance-trained athletes could increase the total glycogen concentration during both of the trials: (1) when only CHO increased (from 6.1 g·kg^−1^.day^−1^ to 7.9 g·kg^−1^.day^−1^) and also (2) when both energy and CHO increased (+34% of energy and 10.5 g CHO·kg^−1^·day^−1^).

In this line, a study with nine athletic women, McLay et al. [14] submitted to 3 days of CHO loading (8.4 g·kg^−1^·day^−1^ CHO) and 3 days of an isoenergetic normal diet (5.2 g·kg^−1^·day^−1^ CHO) showed a significant increase (27%) in the resting muscle glycogen concentration in the mid FP, but not in the mid-LP. However, performance was not affected by diet or MC phase, and the MC phase had no effect on substrate utilization during exercise. In another study with six well-trained athletic women in the mid LP [120], muscle glycogen concentration showed only a modest increase (13%) with a CHO loading dose around 8.2 g·kg·day^−1^, compared with a moderate CHO diet of around 4.7 g CHO g·kg·day^−1^. Therefore, the impact of a CHO loading protocol on muscle glycogen concentration seems lower if an athletic woman is in the LP (0%–13%) compared to the FP (17–31%) and compared to athletic men (18–47%) [121].

These findings support previous observations of increased resting muscle glycogen concentration in the mid-LP than mid-FP, proposing that lower glycogen storage in the mid FP can be overcome by CHO loading [12]. Moreover, there is evidence that ingestion of CHO in the LP may help mitigate some of the negative effects observed in this period, such as the decrease in blood glucose to normal levels, as well as helping to support immune function [122]. Taken together, these data suggest that, in athletic women, a lower total daily CHO amount might be needed compared to men; this decrease might be more pronounced in the LP of the MC. Additionally, a CHO loading protocol of >8 g·kg^−1^·day^−1^ might be useful to increase the glycogen concentrations in women in the FP, and an increase in energy intake might be needed to achieve this total daily amount of CHO; however, this protocol might not be needed or, at the very least, a vast difference should not be expected if the athlete is in the mid LP.

#### 2.3.3. Lipids

Women oxidize proportionately more lipids than men at all exercise intensities [123,124]. Additionally, findings suggest that women have a greater content of intramyocellular lipids and a greater capacity to utilize these lipid stores [124,125], with the possibility that sex-based differences are more apparent with increasing exercise training duration [124]. On the other hand, the higher capacity to oxidize lipids during exercise may indicate that women perhaps require lower amounts of CHO during exercise compared to men. Additionally, and considering that estrogen, which is higher in the mid LP, increases fat oxidation and decreases CHO dependence [126], during exercise, an even lower CHO intake, both daily and during exercise, might be considered.

Concomitantly, and as mentioned above, during exercise training, the differences in preferences for energetic substrates [108,115], e.g., fat oxidation and CHO, seem to be related to the levels of estrogen and, eventually, progesterone [12]. However, even when taking into account these differences, there seems to be no advantage to applying certain strategies that aim to take advantage of this increased fat oxidation during exercise performance, such as the ketogenic diet [127]. Furthermore, existing studies do not seem to consider this difference in the oxidation of substrates throughout the MC. Taking all of these data into account, in regard to daily lipid intake, the general population recommendation of 20–35% of the total energy value [128] might be recommended for athletic women.

#### 2.3.4. Protein

Regarding protein, the LP seems to be more catabolic than the FP, with the LP showing a higher leucine flux and oxidation, leading to an increase in resting energy expenditure [129]. Additionally, Sawai et al. [130] also demonstrated that, during the LP, the plasma concentration of several free amino acids was lower compared to the FP, suggesting an accelerated protein catabolism during the LP. There is also evidence of a greater protein catabolism during the LP, with the differences being smaller when a CHO supplement (0.6 g CHO kg·h^−1^, in a total of around 35 g·h^−1^) was ingested during endurance exercise (cycling at 70% peak V̇O_2_ until exhaustion) compared to a placebo drink [131]. This suggests a greater amino acid catabolism at rest and during exercise [16], and that the intake of CHO during exercise might attenuate the amino acid catabolism. Progesterone seems to be the hormone responsible for the increased catabolism in the LP [132], while estrogen may help to reduce protein catabolism [133], both at rest and during exercise [126]. Given this, the need for a greater amount of protein during the LP might be considered [121], along with the intake of protein during exercise. However, it is also possible that these higher levels of catabolism seem to correspond to changes in the amount of protein ingested by athletic women, with an increase in the amount ingested at the end of this phase [134].

A recent systematic review [135] aiming to determine the protein requirements of pre-menopausal (18–45 years) athletic women concluded that the requirements are similar to recreational and/or competitive women undertaking aerobic endurance (1.28–1.63 g·kg^−1^·day^−1^), resistance (1.49 g·kg^−1^·day^−1^) and intermittent exercise (1.41 g·kg^−1^·day^−1^) of ~60–90 min duration. These requirements are aligned with the current sports nutrition guidelines for all athletes (1.2–2.0 g·kg^−1^·day^−1^) [136]. Additionally, since anabolic sensitivity seems to be similar between men and women [137], a similar amount of daily protein might be suggested, with a range of 1.2–2.0 g·kg^−1^·day^−1^ being recommended for the majority of situations regarding athletic performance. Unfortunately, in a systematic review [135], the influence of the MC phase on protein requirements could not be determined. The authors also concluded that 0.32–0.38 g·kg^−1^ pre- and post-exercise demonstrated beneficial physiological responses in recreational and competitive female athletes completing resistance and intermittent exercise. This is within the upper range of the general amount of protein recommended per meal, which is 0.25–0.4 g·kg^−1^·meal^−1^ [138] Box 2.

Box 2Special box—Methodological considerations: Hormonal contraceptive effects on the MC and its impact on sport performance.Combined oral contraceptives (COC) might provide a more controlled and stabilized hormonal profile as they play a dual role: downregulation of endogenous concentrations of estrogen and progesterone, whilst simultaneously providing daily supplementation of exogenous estrogen and progesterone [16,139]. This altered hormonal milieu differs significantly from that of eumenorrheic women and might impact exercise performance due to changes in ovarian hormone-mediated physiological processes [16,139,140]. The endogenous hormonal profile of an COC user, i.e., low levels of estrogen and progesterone, is comparable to the profile observed during the early FP of the MC [141]. COC are commonly used by athletic women, primarily to alleviate symptoms of dysmenorrhea and menorrhagia, reduce the occurrence of premenstrual tension, and other clinical conditions [142]. However, information on menstrual manipulation practices in young physically active women is sparse, with its use being estimated to be similar between athletes and the general population [16,17,143]. The potential side effects of COC impacting performance (or not) are in agreement with the newer progestins, lower dose triphasic pills or continuous pills [139].Multiple variables may be considered in research on sports performance and COC, such as: training status (untrained, trained, elite); number of participants; variety of testing protocols; intensity of the exercise; circadian fluctuation in hormonal secretion; nutrition [16,144]. Physical fitness is generally defined in terms of aerobic and anaerobic power and capacity, as well as muscle strength and flexibility. However, sports performance is a broader and more complex concept. It encompasses neuromuscular, cognitive and psychological functions, with both nature and nurture (genetic factors and training) combined determining athletic prowess [144,145].A meta-analysis from 2020 with 590 participants [141] was the first to used robust assurance tools, aiming to select information from early research that has had its share of methodological inaccuracies. Overall, the use of COC might result in a slightly inferior exercise performance, which may be due to an individual variability in the response of different parameters, particularly changes in substrate metabolism and heat stress response, although any group-level effect was most likely to be trivial [141,146]. However, in elite sports, it must be emphasized that even nonsignificant changes can make the difference in terms of winning a gold medal, and so an individualized approach should be adopted.

## 3. Concluding Remarks

This review has taken rather a long journey, in which the literature has been scrutinized in the search for knowledge regarding the relationships between MC, exercise and nutritional intake in women. Beyond an overview of the mechanisms behind these phenomena, the intention was to also deliver some concrete guidelines for translating theory into practice. In the end, the major take-home message is that relevant interindividual variation exists; therefore, any generic guidelines are prone to a lack of generalizability and may fail to provide much-needed guidance. With regard to sex steroid hormones, we explored the roles of estrogens and progestogens. In both cases, their complex mechanisms are still being unraveled by research. Moreover, their relationships with exercise, either how they impact exercise and/or how exercise impacts their regulation, remain the subject of ongoing research, which has had its fair share of controversy. In general, even the hormonal fluctuations during the phases of the MC cannot be easily correlated with exercise performance. While some internal physiological parameters indeed vary across the MC, their impact on performance seems to be highly variable from woman to woman and the magnitude of effects tends to be residual or trivial at best. The same complex and heterogeneous relationships can be observed between exercise and menopause symptoms. Before the menopause, intake of OCs may at least bring comfort to the women taking them, as they do not have to fear inopportune bleeding and pain, with the impact on exercise performance appearing to be minimal.

The interpretation of research findings concerning energy demands and nutritional intake in women in relation to hormonal fluctuations face similar difficulties, as a higher energy expenditure in some phases of the MC tends to be naturally compensated by an increase in nutritional intake. Therefore, even if an increase in energy is required, it will likely occur naturally. Likewise, strategies aiming to increase fat oxidation do not seem to bring any advantages in terms of exercise performance, and the management of protein intake during the different MC phases is also poorly understood. Overall, this is a promising field of research, but one where the search for populational trends may have to be replaced by highly individualized approaches, due to the considerable heterogeneity and variability of responses.

## Data Availability

Not applicable.

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
