# Peer review of "Bidirectional Interactions between the Menstrual Cycle, Exercise Training, and Macronutrient Intake in Women: A Review"

_nutrients, 2021, doi:10.3390/nu13020438_

Round 1
Reviewer 1 Report
It is an interesting and valuable article, but there are many typos and mistakes such as follows:
L116: As a precursors => As precursors
L138: estradiol, is => Please delete the comma.
L145: Circulating estrogens in the women result of direct ovarian secretion of
estradiol and estrone, and peripheral conversion of its precursors at fat tissue,
skin, muscle, endometrium. => grammatically correct?
L197: Afar its role as a sex hormone, estrogen exert --- => grammatically correct?
L208: effect => effects
L212: stabilize the extracellular => something missing?
L222: In vitro and in vivo skeletal muscle contractility in women ERα skeletal muscle-specific knockout mice and report that force generation is impaired across multiple parameters. => grammatically correct?
L236: In sum => In summary?
L259: Their => The
L284: studies => Please adjust the font size.
L286: women rats => female rats (throughout the manuscript)
L300: LP (LP) => Please delete (LP).
L302: V̇O2max => Please change 2 to small lower letter.
L300, 516: menarche => Please make reword in English.
L562: (HRT9 => (HRT)
L635: male => men (throughout the manuscript)
L656: in the LP but, possibility, also in the FP. => grammatically correct?
L702: Altogether, for the women athlete data suggests that --- => grammatically correct?
L789: its used is similar to => grammatically correct?
L793: The million-dollar question of whether COC’s may help => grammatically correct?
The context is good but some information on the menstrual phase is needed. Please add the contents on the associations among exercise/training, menstrual phase, and performance/conditioning as well in references to such articles as Hayashida, H., et al. Exercise-induced inflammation during different phases of the menstrual cycle. Physiother Rehabil. 2016, 1, 121. doi:10.4172/2573-0312.1000121
Author Response
We would like to thank the careful review of our manuscript. It is our belief that your comments and suggestions certainly enriched the quality of the text and improved substantially the rationale of the discussion. All the suggestions were therefore considered and included in the revised manuscript. They are highlighted by "Track Changes" function.
It is an interesting and valuable article, but there are many typos and mistakes such as follows:
L116: As a precursors => As precursors
Modified in accordance.
L138: estradiol, is => Please delete the comma.
Modified in accordance.
L145: Circulating estrogens in the women result of direct ovarian secretion of
estradiol and estrone, and peripheral conversion of its precursors at fat tissue,
skin, muscle, endometrium. => grammatically correct?
Modified in accordance.
L197: Afar its role as a sex hormone, estrogen exert --- => grammatically correct?
Modified in accordance.
L208: effect => effects
Modified in accordance.
L212: stabilize the extracellular => something missing?
Modified in accordance.
L222: In vitro and in vivo skeletal muscle contractility in women ERα skeletal muscle-specific knockout mice and report that force generation is impaired across multiple parameters. => grammatically correct?
Modified in accordance.
L236: In sum => In summary?
Modified in accordance.
L259: Their => The
Modified in accordance.
L284: studies => Please adjust the font size.
Modified in accordance.
L286: women rats => female rats (throughout the manuscript)
Modified in accordance.
L300: LP (LP) => Please delete (LP).
Modified in accordance.
L302: V̇O2max => Please change 2 to small lower letter.
Modified in accordance.
L300, 516: menarche => Please make reword in English.
Menarche is a proper English word.
L562: (HRT9 => (HRT)
Modified in accordance.
L635: male => men (throughout the manuscript)
Modified in accordance.
L656: in the LP but, possibility, also in the FP. => grammatically correct?
Modified in accordance.
L702: Altogether, for the women athlete data suggests that --- => grammatically correct?
Modified in accordance.
L789: its used is similar to => grammatically correct?
Modified in accordance.
L793: The million-dollar question of whether COC’s may help => grammatically correct?
The sentence was removed to avoid misunderstanding.
The context is good but some information on the menstrual phase is needed. Please add the contents on the associations among exercise/training, menstrual phase, and performance/conditioning as well in references to such articles as Hayashida, H., et al. Exercise-induced inflammation during different phases of the menstrual cycle. Physiother Rehabil. 2016, 1, 121. doi:10.4172/2573-0312.1000121
We have included this study. Please see lines 516-522. Furthermore, wide information regarding the menstrual phases and their relationships with exercise had already been provided throughout this section.
All corrections and suggestions were considered. In the manuscript, the corrections are highlight by "Track Changes" function.

Reviewer 2 Report
This article is a review of training, nutrition and the female sexual cycle. It is an ambitious systematic summary of a vast amount of knowledge, but there are some shortcomings in the description.
Line 133-241 Several articles describe the effect of MC on BCAA and amino acid metabolismas follows:
Mariko Obayashi et al. Estrogen controls branched-chain amino acid catabolism in female rats.
J Nutr 2004 Oct;134(10):2628-33.
https://doi.org/10.1093/jn/134.10.2628
Asuka Sawai et al. The effects of estrogen and progesterone on plasma amino acids levels: evidence from change plasma amino acids levels during the menstrual cycle in women
Biological Rhythm Research 51, 2020 - Issue 1
https://doi.org/10.1080/09291016.2018.1526496
Line 284: The character of "studies" seems not to be bold.
Line 300: The abbreviation of "LP" is not appropriately full spelled at the first appearance in the text.
Line 309: The author's name should be "Sato et al. 2016".
Line 317-326: These articles may contribute to women's performance compared with men in ultramarathon.
Keramida G, Peters AM. Fasting hepatic glucose uptake is higher in men than women. Physiol Rep. 2017;5(11):e13174. doi:10.14814/phy2.13174
Geer EB, Shen W. Gender differences in insulin resistance, body composition, and energy balance. Gend Med. 2009;6 Suppl 1(Suppl 1):60–75. doi:10.1016/j.genm.2009.02.002
Maarten R. Soeters, Hans P. Sauerwein, Johanna E. Groener, Johannes M. Aerts, Mariëtte T. Ackermans, Jan F. C. Glatz, Eric Fliers, Mireille J. Serlie, Gender-Related Differences in the Metabolic Response to Fasting, The Journal of Clinical Endocrinology & Metabolism, Volume 92, Issue 9, 1 September 2007, Pages 3646–3652, https://doi.org/10.1210/jc.2007-0552
Table 1: I am sorry I cannot understand the meanings of Table 1. Layout seems to be wrong or inappropriate. Please confirm.
Line 622: "de" should be the.
Line 638-669: Please unify the reference style of "Tarnopolsky et al" in line 638, 641, and 644).
Author Response
We would like to thank the careful review of our manuscript. It is our belief that your comments and suggestions certainly enriched the quality of the text and improved substantially the rationale of the discussion. All the suggestions were therefore considered and included in the revised manuscript. They are highlighted by "Track Changes" function.
This article is a review of training, nutrition and the female sexual cycle. It is an ambitious systematic summary of a vast amount of knowledge, but there are some shortcomings in the description.
Line 133-241 Several articles describe the effect of MC on BCAA and amino acid metabolism as follows:
Mariko Obayashi et al. Estrogen controls branched-chain amino acid catabolism in female rats.
J Nutr 2004 Oct;134(10):2628-33.
https://doi.org/10.1093/jn/134.10.2628
Asuka Sawai et al. The effects of estrogen and progesterone on plasma amino acids levels: evidence from change plasma amino acids levels during the menstrual cycle in women
Biological Rhythm Research 51, 2020 - Issue 1
https://doi.org/10.1080/09291016.2018.1526496
Modified in accordance. We included both studies. Please see lines 775 and 762—765.
Line 284: The character of "studies" seems not to be bold.
Modified in accordance.
Line 300: The abbreviation of "LP" is not appropriately full spelled at the first appearance in the text.
Modified in accordance.
Line 309: The author's name should be "Sato et al. 2016".
Modified in accordance.
Line 317-326: These articles may contribute to women's performance compared with men in ultramarathon.
Keramida G, Peters AM. Fasting hepatic glucose uptake is higher in men than women. Physiol Rep. 2017;5(11):e13174. doi:10.14814/phy2.13174
We chose not to include this reference, as it includes participants with pathologies, mostly cancer, which may alter the metabolic pathways and substrate utilization.
Geer EB, Shen W. Gender differences in insulin resistance, body composition, and energy balance. Gend Med. 2009;6 Suppl 1(Suppl 1):60–75. doi:10.1016/j.genm.2009.02.002
Added. Please see lines 671-677.
Maarten R. Soeters, Hans P. Sauerwein, Johanna E. Groener, Johannes M. Aerts, Mariëtte T. Ackermans, Jan F. C. Glatz, Eric Fliers, Mireille J. Serlie, Gender-Related Differences in the Metabolic Response to Fasting, The Journal of Clinical Endocrinology & Metabolism, Volume 92, Issue 9, 1 September 2007, Pages 3646–3652, https://doi.org/10.1210/jc.2007-0552
We chose not to include this reference, as it refers to a 38-h fasting, which we believe to be out of scope for our work.
Table 1: I am sorry I cannot understand the meanings of Table 1. Layout seems to be wrong or inappropriate. Please confirm.
We have decided to remove the table, as we believe it was not adding to the overall message of the work.
Line 622: "de" should be the.
Modified in accordance.
Line 638-669: Please unify the reference style of "Tarnopolsky et al" in line 638, 641, and 644).
There are two authors named Tarnopolsky. Unfortunately, while EndNote automatically uses the proper initials to differentiate them in most cases, that does not occur when their name comes in the middle of other authors. We tried fixing through the EndNote records directly but could not change this aspect. However, we believe MPDI will require a change to numbered referencing before publication, and so only numbers will appear, not authors’ names.

Round 2
Reviewer 1 Report
It is recommended to consider to correct following issues:
L147: peripheral conversion of its precursors occur at fat tissue, skin, muscle, endometrium => peripheral conversion of the precursors occurs at fat tissue, skin, muscle, and endometrium
L207: and hence, => , and hence
L223: regulation CHO => regulation of CHO
L226: In vitro and in vivo assays in ERα skeletal muscle-specific knockout mice showed that muscle contractility was impaired. => Please adjust the font size, and ERα might be better to move to between muscle-specific and knockout.
L231: their => the
L291: studies => Please adjust the font size.
L340: and /or => and/or
L303: changes progesterone levels => changes in progesterone levels
L518: calcoprotein => calprotectin
L522: this was more apparent in the menstrual phase => If you like, add the data in https://doi.org/10.3390/sports9010008
L552: Luteinizing => luteinizing
L555: concentrations estradiol => concentrations of estradiol
L661: 90 min => 90-min
L700: were increase => increased
L748, 809: e.g. => e.g.,
L767: -1 => in upper small letters
L768: V̇O2 => 2 should be in a lower small letters
L821: the training status (untrained, trained, elite) number of participants => Don't you need ; between ) and number?
L822: secretion, nutrition) => secretion; nutrition, or secretion, and nutrition? In any case, ) is not needed.
L824: strength, flexibility => strength, and flexibility
Author Response
We would like to thank the careful review of our manuscript. All the suggestions were therefore considered and included in the revised manuscript. They are highlighted by "Track Changes" function.
